# Epidemiological Profile of Ophir Loyola Cancer Hospital: A Snapshot of the Incidence of Solid Neoplasms in the Eastern Amazon

**DOI:** 10.3390/medsci11040068

**Published:** 2023-10-27

**Authors:** Juliana Ramos Chaves, Mateus Itiro Tamazawskas Otake, Diego Di Felipe Ávila Alcantara, Rosilene Silva Lima, Ana Paula Borges de Souza, Janaina Silva da Costa Rodrigues, Margareth Maria Braun Guimarães Imbiriba, Rommel Mario Rodriguez Burbano

**Affiliations:** 1Oncology Research Center, Federal University of Pará, Belém 66073-000, Brazil; julianaramoschaves@gmail.com (J.R.C.); paula.borges.sza@hotmail.com (A.P.B.d.S.); rommel@ufpa.br (R.M.R.B.); 2Hospital Ophir Loyola, Belém 66063-240, Brazil; itirootake@gmail.com (M.I.T.O.); asplanhol@yahoo.com.br (R.S.L.); janainascrodrigues@gmail.com (J.S.d.C.R.); braun.margareth@gmail.com (M.M.B.G.I.)

**Keywords:** cancer, epidemiology, incidence, spatial analysis

## Abstract

Background: Solid neoplasms have a heterogeneous incidence worldwide and in Brazil. Thus, the region delimited by the Legal Amazon has a distinct epidemiological profile. In Pará, Ophir Loyola Cancer Hospital(OLCH) accounts for 71.11% of hospital visits in the state. Methods: This was an ecological, exploratory, and mixed descriptive studythat investigated the epidemiological profile of patients with cancer treated at OLCH from January to December 2020. Sociodemographic data at admission were the primary variables, which were analyzed according to spatial distribution. Results: In this study, the data of 2952 patients were analyzed, with the majority being between the ages of 50 and 79 years (62.47%), female (59.49%), and diagnosed but without previous treatment (87.30%). The most common cancers were breast (16.50%), cervical (13.40%), stomach (8.98%), and prostate (7.72%). Of the 12 integration regions, Guajará had the highest number of referrals (49.86%), followed by Guamá (12.94%) and Caeté River (8.98%). Conclusion: The profile of care at OLCH showed a high incidence of solid malignancies compared to that in other regions of Brazil, indicating environmental and sociocultural influences on the carcinogenic profile present in the eastern Amazon.

## 1. Introduction

The geographical distribution of cancer is a major concern in global public health [1]. Associated with this reality, the susceptibility of a population is closely linked to risk factors and carcinogenic profiles in different groups, which explain the heterogeneous prevalence and incidences among countries [2,3].

Brazil, compared to other regions of the world, has a distinct epidemiological profile, with a high prevalence of breast cancer and a low survival rate compared to those for countries with a higher human development index [4,5]. In addition, the North and Northeast regions account for more than half of the cases of penile cancer, with a high mortality rate [6], as well as prostate cancer, which is the second most commonly diagnosed cancer in men over 40 years of age [7,8].

Regarding the proportion of cases, in the North Region, the incidence of stomach cancer ranks second among men and third among women [9], with mortality rates greater than 11.2 and 2.4 per 100 thousand inhabitants among men and among women, respectively, aged 50–69 years [10].

In an attempt to explain this profile among Brazilian regions, miscegenation is a relevant characteristic of the population of the State of Pará. Miscegenation occurs when populations that have remained isolated for hundreds of generations come together in a geographic space, and individuals from different populations of origin marry and reproduce [11]. Studies have indicated that the genetic aggregation of Asian and Caucasian population groups with the Amerindian population residing in Brazil, which over the years initiated this process of miscegenation, potentiated the introduction of genotypes favorable to certain malignancies [12,13].

In Brazil, highly complex units and centers of the Unified Health System are responsible for the majority of care for cancer patients [14]. In this context, the Ophir Loyola Cancer Hospital (OLCH) is considered a high-complexity oncology center (CACON) because it has the technical conditions, physical facilities, equipment, and human resources adequate to provide highly complex specialized care for the diagnosis and treatment of all types of cancer.

Due to the specific incidence and prevalence profile of cancers in the North Region, compared to those for the national scenario, the need to understand the regional epidemiological characteristics is associated with, as a priority, the identification of the main solid tumors. Thus, the objective of this study was to describe the epidemiological profile of patients treated in the CACON of OLCH who had confirmed diagnoses of solid neoplasms in 2020.

## 2. Method

This was an ecological, exploratory, and mixed descriptive study for which the reported and registered data of 2952 patients in the state of Pará and other states who were diagnosed and treated at the CACON of OLCH were analyzed. Data for sociodemographic, diagnostic, and region of origin variables were obtained from the hospital cancer registries (HCR), available in the information system maintained by the National Cancer Institute (NCI).

The data were grouped by integration region (IR) in Pará: Guajará, Guamá, Marajó, Caeté River, Capim River, Tocantins, Araguaia, Tucuruí Lake, Carajás, Xingu, Lower Amazon, and Tapajós. To illustrate the incidence and prevalence of different cancers in the integration regions of Pará, maps containing the regional distribution of registered cases were constructed.

The analysis and grouping of primary tumors wereconducted with reference to the International Statistical Classification of Diseases and Related Health Problems (ICD), 11th Revision, from code C00 to C97, and the International Classification of Diseases for Oncology (ICD-O), 3rd Edition. The identified, recorded, and organized information underwent descriptive statistical analyses, i.e., measures of central tendency (mean, mode, and median), variance and standard deviation, and absolute and relative frequencies, using IBM SPSS Statistics 27 and GraphPad Prism 6 software. A *p*-value ≤ 0.05 was considered significant.

## 3. Results

In 2020, there were 2952 patient admissions for cancer at OLCH, which is located in the Guajará integration region. Of this total, 1196 (40.5%) were men, and 1756 (59.5%) were women; 62.47% were between 50 and 79 years of age, and 31.06% were between 20 and 49 years of age. Furthermore, around 2577 (87.30%) patients began follow-up with the diagnosis, but without starting the first therapeutic modality. Table 1 shows the number of cases by age group, sex, education level, presence or absence of diagnosis and treatment prior to follow-up, and race or ethnicity.

The numbers and percentages of cancer cases confirmed and recorded by OLCH are shown in Table 2, compared with the number of cases in Pará state and Brazil, in 2020.

The numbers and percentages of cancer cases confirmed and recorded by OLCH are shown in Table 3, compared with the number of cases in Brazil, in 2020.

The spatial distribution of the total number of cases, by the location of the primary solid tumor, was heterogeneous with regard to the number of cases per municipality and per integration region (Figure 1). The municipality of Belém had the highest number of cases (34.01%), followed by the municipalities of Nonindium (11.62%), Castanhal (4.51%), Abaetetuba (2.47%), Bragança and Marituba (2.47%), Paragominas (1.69%), and Capanema (1.63%). Guajará accounted for 49.86% of the registered cases, with Marajó, Tocantins, Capim River, Caeté River, and Guamá accounting for 43.77% (1343 cases); Lower Amazon, Tucuruí Lake, Carajás, Araguaia and Xingu accounted for 5.72% (169 cases), with other states accounting for 0.64% (Table 4 and Table 5).

The main solid tumors also had a heterogeneous incidence profile in the population of Pará (Table 4 and Table 5). Breast cancer was the most common cancer, which occurred in 489 patients (16.57%), followed by cervical cancer, which occurred in 396 patients (13.41%), stomach cancer, which occurred in 265 patients (8.98%), prostate cancer, which occurred in 225 patients (7.62%),cancer of the colon and rectum, which occurred in 191 patients (6.47%), leukemias, which occurred in 120 patients (4.07%), lung cancer, which occurred in 92 patients (3.12%) and lymphomas, without a specific location, which occurred in 87 patients (2.95%). Other neoplasms accounted for 36.81% of the total number of cases.

Cancer within the digestive tract was the most common, with 624 cases in 2020 [265 (42.47%) cases of stomach cancer, 191 (30.61%) cases of colon and rectum cancer, 65 (10.41%) cases of cancer affecting the liver and intrahepatic bile ducts as well as the gallbladder and extrahepatic bile ducts, and 47 (7.53%) cases affecting the pancreas]; 8.98% of cases were distributed among the esophagus and small intestine and in unspecified locations.

Cervical cancer accounted for 396 (78.42%) of 505 cases of gynecological cancers, followed by ovarian cancer (51 cases, 10.10%) and uterine body cancer (43 cases, 8.51%). In addition, when mastology was included within this group, breast cancer accounted for 49.20% of the total registered cases, bringing the total number to 994 cases.

For solid tumors located in the head and neck, there were 254 cases [108 (42.52%) cases of thyroid cancer, 81 (31.89%) cases of lip and oral cavity cancer, and 33 (12.99%) cases oflarynx cancer]. The remaining cancers (nasal cavity and middle ear, pharynx, nasopharynx, paranasal sinuses, and ocular) accounted for 12.96% of the cases. Finally, there were 386 cases of urological cancer [225 (58.29%) cases of prostate cancer; 75 (19.43%) cases of kidney and renal cancer, 37 (9.59%) cases of bladder cancer, 32 (8.29%) cases of penile cancer, and 17 (4.40%) cases of testicular cancer] (Figure 2).

Within the relationship between females and males, except for cervical, breast, and prostate cancers and excluding non-melanoma skin cancer, there was a higher proportion of cases of stomach cancer among men in all IRs. However, the highest observed number ofthyroid cancer cases occurred among women (Table 3).

## 4. Discussion

The results of this study indicate the importance of OLCH as the reference center in Pará for the care and treatment of cancer patients. Based on the sociodemographic data analyzed, there was a high incidence of cases among patients between 20 and 49 years of age (31.06%), with the majority being female (59.49%). Additionally, approximately 87.30% of the people referred to OLCH had a previous diagnosis but had not started treatment, and only 12.64% had a previous diagnosis and had started the first therapeutic modality.

In terms of spatial representation, Pará encompasses 1,253,164.5 km^2^, making it the second largest state in Brazil in territorial extension, 35.54% larger than the Southeast Region (924,558,341 km^2^), and larger than countries such as France, Spain, and Ukraine. In this regard, the Legal Amazon accounts for approximately 24.9% of the total extension (5,016,478.27 km^2^) of Pará as well as 30.9% of its resident population (28,419,712 inhabitants), thus being the second most populous IR [15]. Thus, given its importance in area and population, the flow of patients to OLCH is noteworthy.

To identify the distribution of cancer cases, Figure 3 details the access protocol of the medium and high-complexity assistance network in oncology of the State of Pará (2021) [16], which directs case flow based on the macro-region and type of cancer. Case flow is determined by institution availability, confirmed diagnosis or clinical suspicion of cancer, and tumor profile, leading to referrals to the high-complexity care units and high-complexity center at OLCH [16].

However, due to the limited number of beds and thus the ability to treat a reduced number of patients with cancers at each UNACON, CACON/OLCH has become the reference institution for patients who need a hospital bed, a lower cost of care, and radiotherapy as well as for patients who have cancers that are not treated at certain UNACONs. Of the estimated number of cancer cases in Pará for 2020 [17], the percentage recorded in the INCA hospital cancer registries (HCR/INCA) for OLCH accounted for approximately 71.11% of all patients in the state, corroborating the high demand observed [16].

In addition to this analysis, the comparison of this percentage of patients who begin follow-up at OLCH with the total registered in Brazil in 2020, indicates a strong prevalence of preventable neoplasms. However, the flow of oncology patients to high-complexity centers in 2020, compared to 2019, wasdrastically reduced, following the manifestation of the first waves of the COVID-19 pandemic, by approximately 45% to 66% [18,19]. Furthermore, the primary tumors with the highest percentage of reduction in diagnosis were colorectal, prostate, and bladder [20].

The coherence found that justifies this reality is directly associated with the re-formulation of surveillance plans and technical support for health services, in accordance with the need for isolation and reduction of the risk of contagion [21]. Furthermore, the findings found in the 2020 records at OLCH suggest the need for an epidemiological description of the reality of a large oncology care center in the eastern Amazon region.

Figure 1 illustrates the spatial distribution of cases by municipality and the percentage by IR, representing the flow of referrals to OLCH from 119 municipalities in Pará and fivein other states. In this context, the highest flows originated in Guajará (49.86%) and Guamá (12.94%). Furthermore, as seen in Figure 2, the distribution of malignant neoplasms in the state followed the prevalence of cancer in each region of Brazil, with breast, cervical, stomach, and colorectal cancers being the most prominent.

In several regions of Brazil, breast cancer overlaps with cervical cancer or has a similar profile [22]. However, in patients treated at CACON/OLCH, for example, from some integration regions, the proportions of these cancers were different (Figure 2, Table 4 and Table 5). In the state of Pará, breast cancer is the most common and causes the highest mortality, which is different from the national data where it may occupy the second or third position (Table 2).

Stomach cancer was also prevalent among patients treated at CACON/OLCH (Figure 2, Table 4 and Table 5). An association between the regional culture of excessive consumption of foods with a hypersodium diet and the high incidence and mortality in Paráhas already been scientifically established [23,24,25] and is not observed in other regions and other states [26] (Table 2 and Table 3). This evidence explains the proportion of cases in the Salgado micro-region, which is composed of the municipalities of the Guamá, Caeté River, and Guajará IRs in northeastern Pará (Figure 2, Table 4 and Table 5). Thus, the epigenetic favoring of mutated genes present in gastric cancer caused by a high-sodium diet is closely associated with the prevalence of this neoplasm in Pará [24].

In Tucuruí Lake and Carajás, thyroid cancer was the second most frequent neoplasms, and in the Lower Amazon, it was the third most frequent among patients treated at OLCH (Figure 2, Table 4 and Table 5). In contrast, this neoplasm only ranks asnineteenth on the national scene. Previous studies have indicated the high mining activity in Carajás and the Lower Amazon with a strong relationship between methyl mercury and mercury contamination in the Tapajós River and the adjacent gold extraction territory and the high load of these materials in fish distributed regionally for human consumption [27,28,29]. In addition, exposure to mercury increases the risk of developing thyroid cancer [30].

Of the total solid tumors identified in the 2952 patients, the 10 most incident tumors (except non-melanoma skin cancer) were breast cancer, cervical cancer, stomach cancer, prostate cancer, colon and rectum cancer, leukemia (bone marrow), thyroid cancer, lung cancer, lymphoma (unspecified), and lip and oral cavity cancer.

Breast cancer accounted for the highest number of cases (Table 2), corroborating its high prevalence in the northern regions in an ecological study conducted by Camargo et al. (2021) [31]. In the data recorded by CACON, approximately 35.79% of cases were in patients aged between 20 and 49 years, 24.95% were in patients aged between 40 and 49 years, and 64.21% were in patients over 50 years of age. However, in recent years, there has been a trend in Brazil toward an increase in the proportion of cases in patients under 40 years of age [32].

The increase in this frequency and the predominance in areas with greater human and socioeconomic development have been associated with greater access to diagnostic and detection tools, as exemplified by the high prevalence of advanced-stage disease (III and IV) in women under 50 years of age [33]. However, the high number of cases is also associated with low preventive coverage, with a deficit in monitoring and offering screening tests in more precarious regions [32].

On a global level, cervical cancer is the fourth most prevalent cancer among all types and the second most prevalent among females, in 2020 reaching an estimated 604 thousand cases and 342 thousand deaths [34]. In Brazil, cervical cancer is the third most common disease in women, especially in the Centra-West Region [32]. However, in Pará, the incidence of this neoplasm ranked second among women treated at OLCH (Table 1 and Table 2), with 48.23% being between 20 and 49 years, 48.74% being older than 50 years and only 3.03% being younger than 20 years.

In reference to this reality, although screening and detection tests are available for this malignancy, in the last decade there was a significant decline in the coverage of preventive tests, such as cytopathological and histopathological tests, in regions of Brazil, especially in the North, Southeast and South, in addition to an increase in the time it takes to start treatment [35,36]. Additionally, the North Region has one of the highest rates of cervical cancer in Brazil but does not have the same prevalence of confirmed cases of HPV [37]. Thus, the decline in early detection strategies and the limited vaccination coverage in the last 3 years may be closely linked to the incidence of this neoplasia in Pará.

Gastric cancer is a public health problem affecting onein 54 men and onein 126 women worldwide [38]. In Brazil, stomach cancer ranks fourth among male malignancies, mainly affecting those aged between 60 years and 74 years [26]. In the general Brazilian population, in 2020, this primary tumor location ranked eighth (Table 2), and there were approximately 13,360 new cases among men and 7870 among women in the period from 2020 to 2022 [17].

Among the Brazilian regions, for males, gastric cancer is more common in the South (16.02/100,000), Southeast (13.99/100,000), and North (11.75/100,000) Regions [17]. At CACON/OLCH, the incidence of gastric cancer ranks third, with 265 cases, with a predominance of cases in males (67.55%) than in females (32.45%); 24.15% of patients with these neoplasms were younger than 50 years, with a higher incidence among those between 50 and 69 years of age (49.05%).

As the second most common malignancy among men worldwide, prostate cancer accounts for an estimated 65,000 cases in Brazil alone, being the second most lethal cancer in this population [39]. Of the 225 patients treated at OLCH, the majority were over 70 years of age (51.55%), and 46.67% were between 50 and 69 years of age; prevalence rates that are close to the national rate [39], which for 2020 was 33.08/100,000 inhabitants, approximately 3 times higher than that for lung cancer and trachea cancer, thus contradicting the global trend, which is the opposite [17].

Malignant neoplasms of the colon and rectum were the fourth most frequent and the third most frequent cancers among females at OLCH (Table 2). Worldwide, the incidence of these neoplasms ranks second, with the fifth-highest mortality rate among men [34]. In Brazil, the estimated rate of new diagnoses per year between 2020 and 2022 was 20,540 for males and 20,470 for females, associated with an increased influence of modifiable risk factors [40,41]; these values differ from the proportions at CACON/OLCH, with malignant neoplasms affecting more women (53.40%) than men (46.60%) in 2020 [42]. Therefore, in this study, of the total number of cases with a diagnosis or clinical suspicion of colorectal cancer, 39.27% were between 40 and 59 years of age, 49.21% were older than 60 years, and 11.52% were between 20 and 39 years of age.

Furthermore, some sociocultural factors suggest explanations directly related to the different incidences of this neoplasia in the IRs of Pará (Table 4 and Table 5). Recent studies indicate that the coverage of specialized diagnostic services and preventive follow-up for colorectal cancer are poorly distributed between the interior and the capital of the states in the North Region, in addition to the rapid growth, in the last decade, of sedentary lifestyles and obesity [43,44].

The global epidemiological profile of thyroid cancer indicates a higher incidence in females, with an estimated risk of 3 to 10 times greater than that for males (Table 3), depending on the region [45]. In Brazil, for this malignancy, the estimate for 2020 was 13,780 cases, with a higher prevalence in the South and Southeast Regions [17].

Based on the local data from OLCH, thyroid cancer, in general, was the seventh most frequent, excluding non-melanoma skin cancer (Table 4 and Table 5), with 86.11% of cases in females and only 13.89% in males. In addition, 39.82% of those with thyroid cancer were between 20 and 39 years of age, 44.44% were between 40 and 59 years of age, and 15.74% were older than 60 years. In recent years, there has been a significant increase in the incidence of these neoplasms in the population due to the carcinogenic profile itself and advances in diagnosis, which have allowed the early detection of subtypes of differentiated papillary, follicular, and Hürthle cell carcinomas [34,46].

## 5. Conclusions

Considering the admissions and treatments to CACON/OLCH for 2020, there were high incidences of breast, cervical, stomach, prostate, colon, and rectal cancersin individuals with an average age close to that expected based on the national average. However, the disparities in the incidence rates and the growth in the number of less expressive types of cancer show that the carcinogenic profile and the influence of environmental factors specific to each IR promote a specific epidemiological profile for the eastern Amazon.

In addition, Guajará stands out as the IR with the highest flow of referrals, and Marajó has the highest proportion of patients with cancer who underwent screening and the lowest coverage of preventive follow-up.

## Figures and Tables

**Figure 1 medsci-11-00068-f001:**
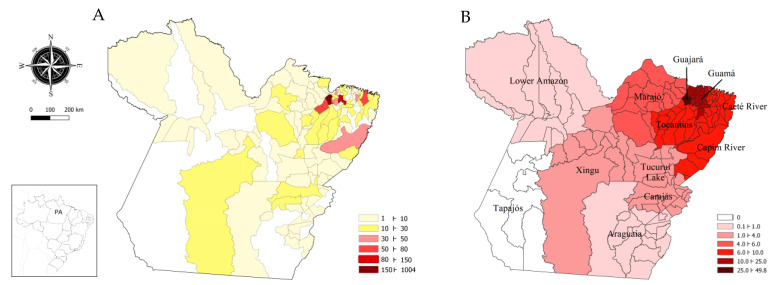
Spatial distribution of cases treated at OLCH in 2020. (**A**) Absolute number of cases per municipality. (**B**) Percentage of cases by integration region of Pará.

**Figure 2 medsci-11-00068-f002:**
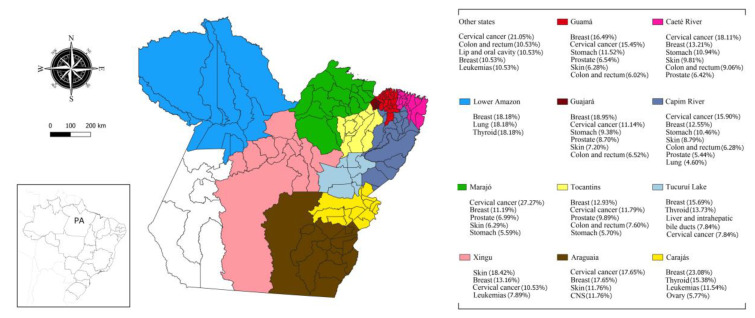
Percentage of the most frequent cancers by integration region.

**Figure 3 medsci-11-00068-f003:**
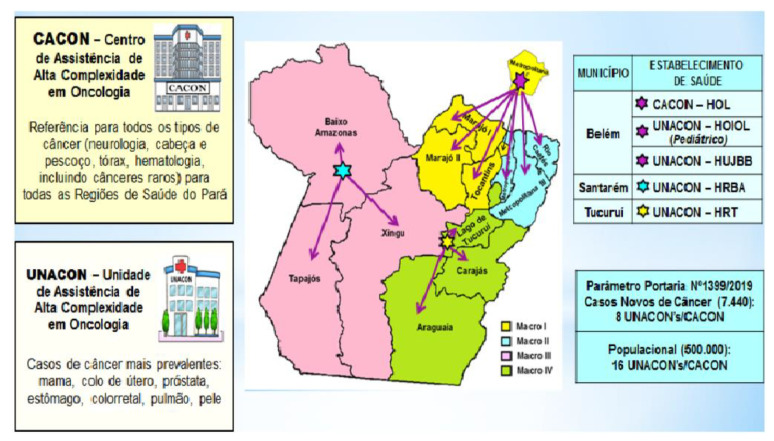
Flow of patients to high-complexity care centers and units. Source: adapted from the access protocol of the medium and high-complexity assistance network in oncology of the State of Pará, 2020.

**Table 1 medsci-11-00068-t001:** Epidemiological description of patients with solid tumors in Pará treated at OLCH in 2020.

Variables	*n* (%)
Age group	
≤19 years	12 (0.41)
20–49 years	917 (31.06)
50–79 years	1844 (62.47)
≥80 years	179 (6.06)
Sex	
Male	1196 (40.51)
Female	1756 (59.49)
Education level	
None or <1 year	238 (8.06)
<8 years	1172 (39.70)
8–10 years	264 (8.94)
11–14 years	640 (21.68)
>15 years	155 (6.47)
Not reported	447 (15.14)
Previous diagnosis/treatment	
With diagnosis/without treatment	2577 (87.30)
With diagnosis/with treatment	373 (12.64)
Not reported	2 (0.06)
Color/ethnicity	
Yellow	183 (6.20)
Brown	1865 (63.18)
Black	52 (1.76)
White	85 (2.88)
Indigenous	4 (0.14)
Not reported	763 (25.85)

**Table 2 medsci-11-00068-t002:** Detailed breakdown of the number of cancer cases by primary tumor location in Pará (OLCH) in 2020.

Primary Tumor Location	OLCH	Pará	Brazil
	n (%)	n (%)	n (%)
Breast	489 (16.5)	646 (15.56)	51,025 (11.96)
Cervix	396 (13.4)	532 (12.82)	25,683 (6.02)
Stomach	265 (8.98)	338 (8.14)	16,680 (3.91)
Prostate	225 (7.72)	312 (7.52)	32,681 (7.66)
Non-melanoma skin	214 (7.25)	373 (8.99)	58,306 (13.67)
Colon and rectum	191 (6.47)	251 (6.05)	36,426 (8.54)
Leukemia, bone marrow	120 (4.07)	178 (4.29)	7236 (1.70)
Thyroid	108 (3.66)	134 (3.23)	6526 (1.53)
Lungs	92 (3.12)	128 (3.08)	12,755 (2.99)
Lymphoma, unspecified	87 (2.95)	122 (2.94)	19,464 (4.56)
Lip and oral cavity	81 (2.74)	98 (2.36)	12,975 (3.04)
Kidney	75 (2.54)	90 (2.17)	5001 (1.17)
Bones and soft tissues	67 (2.27)	120 (2.89)	15,493 (3.63)
CNS	54 (1.83)	94 (2.26)	6570 (1.54)
Ovary	51 (1.73)	70 (1.69)	6593 (1.55)
Liver and intrahepatic bile ducts	50 (1.69)	63 (1.52)	3292 (0.77)
Pancreas	47 (1.59)	65 (1.57)	4222 (0.99)
Body of uterus	43 (1.46)	53 (1.28)	7244 (1.70)
Esophagus	38 (1.29)	67 (1.61)	7794 (1.83)
Bladder	37 (1.25)	49 (1.18)	7365 (1.73)
Larynx	33 (1.12)	44 (1.06)	5383 (1.26)
Penis	32 (1.08)	44 (1.06)	1162 (0.27)
Not specified	30 (1.02)	59 (1.42)	48,056 (11.27)
Hematopoietic and reticulo endothelial systems, not bone marrow	23 (0.78)	49 (1.18)	3383 (0.79)
Testicle	17 (0.58)	27 (0.65)	1873 (0.44)
Gallbladder and extrahepatic bile ducts	15 (0.51)	19 (0.46)	2356 (0.55)
Small intestine	10 (0.34)	11 (0.26)	1863 (0.44)
Nasopharynx	10 (0.34)	12 (0.29)	1097 (0.26)
Vulva	8 (0.27)	13 (0.31)	2375 (0.56)
Digestive system, unspecified	8 (0.27)	8 (0.19)	3629 (0.85)
Ocular	7 (0.24)	18 (0.43)	1035 (0.24)
Pharynx	6 (0.20)	31 (0.75)	7114 (1.67)
Mediastinum	5 (0.17)	7 (0.17)	1276 (0.30)
Nasal cavity and middle ear	5 (0.17)	5 (0.12)	629 (0.15)
Paranasal sinuses	4 (0.14)	6 (0.14)	810 (0.19)
Vagina	4 (0.14)	8 (0.19)	554 (0.15)
Placenta	3 (0.10)	3 (0.07)	307 (0.07)
Respiratory system, unspecified	1 (0.03)	2 (0.05)	131 (0.03)
Thymus	1 (0.03)	2 (0.05)	145 (0.03)
Total	2952 (100.00)	4151 (100.00)	426,509 (100.00)

**Table 3 medsci-11-00068-t003:** Distribution of cases per sex (by primary tumor location) treated at OLCH and Brazil, in 2020.

Primary Tumor Location	OLCH n (%)		Brazil n (%)	
	Males	Females	Males	Females
Stomach	179 (67.55)	86 (32.45)	8799 (52.75)	7881 (47.25)
Non-melanoma skin	116 (54.21)	98 (45.79)	27,827 (76.39)	30,479 (23.61)
Colon and rectum	89 (46.60)	102 (53.40)	17,965 (49.32)	18,461 (50.68)
Leukemia, bone marrow	69 (57.50)	51 (42.50)	4033 (55.74)	3203 (44.26)
Thyroid	15 (13.89)	93 (86.11)	1027 (15.74)	5499 (84.26)
Lungs	50 (54.35)	42 (45.65)	6950 (54.49)	5805 (45.51)
Lymphoma, unspecified	53 (60.92)	34 (39.08)	10,170 (52.25)	9294 (47.75)
Lip and oral cavity	50 (61.73)	31 (38.27)	8992 (69.30)	3983 (30.70)
Kidney	43 (57.33)	32 (42.67)	2856 (57.11)	2145 (42.89)
Bones and soft tissues	27 (40.30)	40 (59.70)	7747 (50.00)	7746 (50.00)
CNS	31 (57.41)	23 (42.59)	3433 (52.25)	3137 (47.75)
Liver and intrahepatic bile ducts	26 (52.00)	24 (48.00)	1710 (51.94)	1582 (48.06)
Pancreas	26 (55.32)	21 (44.68)	2098 (49.69)	2124 (50.31)
Esophagus	23 (60.53)	15 (39.47)	5457 (70.02)	2337 (29.98)
Bladder	30 (81.08)	7 (18.92)	5116 (69.63)	2231 (30.37)
Larynx	25 (75.76)	8 (24.24)	4610 (85.64)	773 (14.36)
Hematopoietic and reticuloendothelial systems, not bone marrow	11 (47.83)	12 (52.17)	1506 (44.52)	1877 (55.48)
Gallbladder and extrahepatic bile ducts	4 (26.67)	11 (73.33)	929 (39.43)	1427 (60.57)
Small intestine	4 (40.00)	6 (60.00)	910 (48.85)	953 (51.15)
Nasopharynx	4 (40.00)	6 (60.00)	768 (70.01)	329 (29.99)
Digestive system, unspecified	4 (50.00)	4 (50.00)	1662 (45.80)	1967 (54.20)
Ocular	6 (85.71)	1 (14.29)	569 (54.98)	466 (45.02)
Pharynx	4 (66.67)	2 (33.33)	5903 (82.98)	1211 (17.02)
Mediastinum	3 (60.00)	2 (40.00)	597 (46.79)	679 (53.21)
Nasal cavity and middle ear	2 (40.00)	3 (60.00)	398 (63.28)	231 (36.72)
Paranasal sinuses	2 (50.00)	2 (50.00)	456 (56.30)	354 (43.70)
Respiratory system, unspecified	1 (100.00)	-	69 (52.67)	62 (47.33)
Thymus	-	1 (100.00)	72 (49.66)	73 (50.34)

**Table 4 medsci-11-00068-t004:** Distribution of the incidence of principal solid tumors by location (ICD-11 and ICD-O) and sex in Guajará, Guamá, Caeté River, Capim River, Marajó, and Tocantins in 2020.

Location (ICD-O)	Female (%)	Male (%)
Guajará	864 (100)	608 (100)
Stomach (C16)	52 (6.02)	86 (14.14)
Colon and rectum (C18–C21)	48 (5.56)	48 (7.9)
Bone marrow leukemia (C42.1)	36 (4.17)	41 (6.74)
Breast (C50)	279 (32.29)	-
Cervix (C53)	164 (18.98)	-
Prostate (C61)	-	128 (21.05)
Thyroid (C73)	39 (4.51)	8 (1.32)
Guamá	239 (100)	143 (100)
Stomach (C16)	13 (5.44)	31 (21.68)
Colon and rectum (C18–C21)	13 (5.44)	10 (7.0)
Breast (C50)	61 (25.52)	2 (1.4)
Cervix (C53)	59 (24.69)	-
Prostate (C61)	-	25 (17.48)
Thyroid (C73)	11 (4.6)	2 (1.4)
Caeté River	162 (100)	103 (100)
Stomach (C16)	7 (4.32)	22 (21.36)
Colon and rectum (C18–C21)	16 (9.87)	8 (7.76)
Breast (C50)	34 (20.99)	1 (4.0)
Cervix (C53)	48 (29.63)	-
Prostate (C61)	-	17 (16.50)
Thyroid (C73)	5 (3.09)	-
Lymphoma (C77)	2 (1.23)	5 (4.85)
Capim River	148 (100)	91 (100)
Stomach (C16)	8 (5.41)	17 (18.68)
Colon and rectum (C18–C21)	8 (5.41)	7 (7.7)
Breast (C50)	29 (19.59)	1 (1.10)
Cervix (C53)	38 (25.68)	-
Prostate (C61)	-	13 (14.29)
Thyroid (C73)	10 (6.76)	-
Marajó	82 (100)	61 (100)
Stomach (C16)	-	8 (13.11)
Breast (C50)	16 (19.51)	1 (4.0)
Cervix (C53)	39 (47.56)	-
Prostate (C61)	-	10 (16.39)
Thyroid (C73)	6 (7.32)	-
Lymphoma (C77)	2 (2.44)	5 (8.20)
Tocantins	148 (100)	115 (100)
Stomach (C16)	3 (2.03)	12 (10.43)
Colon and rectum (C18–C21)	13 (8.78)	7 (6.09)
Breast (C50)	34 (22.97)	-
Cervix (C53)	31 (20.95)	-
Prostate (C61)	-	26 (22.61)
Thyroid (C73)	8 (5.41)	-

**Table 5 medsci-11-00068-t005:** Distribution of the incidence of principal solid tumors by location (ICD-11 and ICD-O) and sex in Tucuruí Lake, Carajás, Xingu, Araguaia, and Lower Amazon in 2020.

Topography (ICD-O)	Female (%)	Male (%)
Tucuruí Lake	26 (100)	25 (100)
Stomach (C16)	1 (3.85)	1 (4.0)
Liver and intrahepatic bile ducts (C22)	1 (3.85)	3 (12.0)
Breast (C50)	7 (26.92)	1 (4.0)
Cervix (C53)	4 (15.38)	-
Thyroid (C73)	6 (23.08)	1 (4.0)
Carajás	36 (100)	16 (100)
Stomach (C16)	1 (2.78)	1 (6.25)
Larynx (C32)	-	2 (12.5)
Lung (C34)	1 (2.78)	1 (6.25)
Bone marrow leukemia (C42.1)	3 (8.33)	4 (25.0)
Breast (C50)	12 (33.33)	-
Cervix (C53)	3 (8.33)	-
Thyroid (C73)	6 (16.67)	2 (12.5)
Xingu	19 (100)	19 (100)
Bone marrow leukemia (C42.1)	-	4 (21.05)
Breast (C50)	5 (26.32)	-
Cervix (C53)	4 (21.05)	-
Prostate (C61)	-	2 (10.53)
Araguaia	11 (100)	6 (100)
Bone marrow leukemia (C42.1)	-	1 (16.67)
Breast (C50)	2 (18.18)	1 (16.67)
Cervix (C53)	3 (27.27)	-
Prostate (C61)	-	1 (16.67)
Frontal lobe of the CNS (C71)	2 (18.18)	-
Lower Amazon	7 (100)	4 (100)
Stomach (C16)	1 (14.29)	-
Lung (C34)	2 (28.57)	-
Breast (C50)	2 (28.57)	-
Cervix (C53)	1 (14.29)	-
Prostate (C61)	-	1 (25.0)
Thyroid (C73)	-	2 (50.0)

## Data Availability

Not applicable.

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
