# Peer review of "Epidemiological Profile of Ophir Loyola Cancer Hospital: A Snapshot of the Incidence of Solid Neoplasms in the Eastern Amazon"

_medsci, 2023, doi:10.3390/medsci11040068_

Round 1

Reviewer 1 Report

This is a descriptive ecological study concerning a specific reference center in Brazil, the Ophir Loyola Cancer Hospital (OLCH) in Para, which however covers a very large area of the country, more extensive than many large European countries. I think it would be more interesting to publish it in a Brazilian national journal rather than an international one. For the publication, and if the Editor of the Medical Sciences decides to do so, I have the following observations.

1) The presentation through maps of the incidence of solid tumors in the different areas covered by the hospital is excellent. I don't think anything else is needed at the cartography level.

2) The tables presented in the article need comparative data from Brazilian national data (or global one). E.g. Table 2 which gives the distribution by the primary tumor location should include the corresponding Brazilian national data (even the global one). What was, for example, the incidence of breast or cervix cancer (in absolute and relative frequencies) as a whole throughout Brazil in 2020?

3) The same applies to Tables 6 and 7, where the tumors distribution is specified by gender. These tables must include the national incidence data (or even global data) specified by gender and tumor location .

4) Based on the analytical differences with national or global data, comparisons should be made and the possible interpretations of the differences should be mentioned. The risk factors that are prevalent in the specific area or, conversely, the protective factors that exist against a tumor type must be mentioned. (E.g. what is the interpretation given by the authors on the differences of colon or gastric cancer with the global incidence of these cancer types. Is it the special dietary habits of the region for example?).

Otherwise I think the article is well written, but it needs to be modified regarding the tables I mentioned.

Author Response

Reviewer 1

This is a descriptive ecological study concerning a specific reference center in Brazil, the Ophir Loyola Cancer Hospital (OLCH) in Para, which however covers a very large area of the country, more extensive than many large European countries. I think it would be more interesting to publish it in a Brazilian national journal rather than an international one. For the publication, and if the Editor of the Medical Sciences decides to do so, I have the following observations.

1) The presentation through maps of the incidence of solid tumors in the different areas covered by the hospital is excellent. I don't think anything else is needed at the cartography level.

2) The tables presented in the article need comparative data from Brazilian national data (or global one). E.g. Table 2 which gives the distribution by the primary tumor location should include the corresponding Brazilian national data (even the global one). What was, for example, the incidence of breast or cervix cancer (in absolute and relative frequencies) as a whole throughout Brazil in 2020?

R= We agree with the reviewer's assessment of the analysis. Table 2 indicates the need for an adjusted and ordered overview of the groups of primary solid tumors, from the perspective of OLCH, the state of Pará and Brazil. In this case, in its current form, the size of the numbers recorded by the OLCH would not be clear, without due comparison with the expansion of the territory. Therefore, we grouped data from the states of Pará and Brazil for comparison purposes.

3) The same applies to Tables 6 and 7, where the tumors distribution is specified by gender. These tables must include the national incidence data (or even global data) specified by gender and tumor location .

R= We agree with the reviewer that adding this information to tables 4 and 5 would clarify the extent of the data recorded by OLCH. However, these tables were constructed to visualize the main neoplasms by integration region (IR) stratified by sex. It would not be possible to add national information, as the data in the tables would be confusing and without proper comparability, as well as with an aesthetic deviation that would make understanding difficult. However, in agreement with the reviewer's suggestion and due to the great need for OLCH data to be compared with national data, Table 3 was restructured for this purpose, presenting data by sex of the main groups of malignant neoplasms.

4) Based on the analytical differences with national or global data, comparisons should be made and the possible interpretations of the differences should be mentioned. The risk factors that are prevalent in the specific area or, conversely, the protective factors that exist against a tumor type must be mentioned. (E.g. what is the interpretation given by the authors on the differences of colon or gastric cancer with the global incidence of these cancer types. Is it the special dietary habits of the region for example?).

R= As for the fourth suggestion, we agree with the need to interpret the differences between the quantity registered in the OLCH and at the national level. Therefore, from the 8th paragraph to the 14th paragraph of the discussion, the profile presented by each main malignant neoplasm was interpreted, in order of incidence. For example, the 8th paragraph begins by delimiting the percentage found in the IRs and, above all, the subsequent 11th and 12th paragraphs of the discussion elucidate the main sociodemographic and public health control characteristics related to the factors that suggest a causal effect for this cancer. Thus, it was argued, in the same way, for the other main malignant neoplasms registered in the OLCH.

Otherwise I think the article is well written, but it needs to be modified regarding the tables I mentioned.

Reviewer 2 Report

The authors are describing incidence of various cancers using a registry. Can they please explain how this differs from the cancer incidence within the country? And what is the impact of these differences? Are there any environmental/ epigenetic or social factors that can be attributed to these differences? And how do these differences impact access to care? 

Author Response

Reviewer 2

The authors are describing incidence of various cancers using a registry. Can they please explain how this differs from the cancer incidence within the country? And what is the impact of these differences? Are there any environmental/ epigenetic or social factors that can be attributed to these differences? And how do these differences impact access to care? 

R= As for the suggestion, we agree with the need to interpret the differences between the quantity registered in the OLCH and at the national level. Therefore, from the 8th paragraph to the 14th paragraph of the discussion, the profile presented by each main malignant neoplasm was interpreted, in order of incidence. For example, the 8th paragraph begins by delimiting the percentage found in the IRs and, above all, the subsequent 11th and 12th paragraphs of the discussion, elucidate the main sociodemographic and public health control characteristics related to the factors that suggest a causal effect for this cancer. Thus, it was argued, in the same way, for the other main malignant neoplasms registered in the OLCH.

Reviewer 3 Report

This is a nice study showing distribution and prevalence of solid tumors in the Eastern Amazon. I have no major criticisms.

I only suggest to comment on possible implication of the Covid 19 pandemic on this rates since you have analyzed patients diagnosed and treated in 2020.

Minor editing of English language is required.

Author Response

Reviewer 3

This is a nice study showing distribution and prevalence of solid tumors in the Eastern Amazon. I have no major criticisms.

I only suggest to comment on possible implication of the Covid 19 pandemic on this rates since you have analyzed patients diagnosed and treated in 2020.

R= We appreciate the suggestion and agree with the need to add this influencing factor to hospital records. For this purpose, we added paragraphs 5 and 6 to the discussion chapter, which dialogue with and complement the information present in paragraphs 3 and 4 of the discussion.

Round 2

Reviewer 1 Report

Accepted.